# Investigating microRNA Profiles in Prostate Cancer Bone Metastases and Functional Effects of microRNA-23c and microRNA-4328

**DOI:** 10.3390/cancers15092437

**Published:** 2023-04-24

**Authors:** Helena Järemo, Julius Semenas, Sofia Halin Bergström, Marie Lundholm, Elin Thysell, Anders Widmark, Sead Crnalic, Erik Bovinder Ylitalo, Anders Bergh, Maria Brattsand, Pernilla Wikström

**Affiliations:** 1Department of Medical Biosciences, Pathology, Umeå University, 901 87 Umeå, Sweden; 2Department of Radiation Sciences, Oncology, Umeå University, 901 87 Umeå, Sweden; 3Department of Surgical and Perioperative Sciences, Orthopedics, Umeå University, 901 87 Umeå, Sweden

**Keywords:** prostate cancer, bone metastasis, microarray, proteomics, microRNA-23c, microRNA-4328, proliferation, extracellular vesicles, blood vessels

## Abstract

**Simple Summary:**

Bone metastatic prostate cancer is a lethal disease, and improved treatment strategies are urgently needed. MicroRNAs (miRNAs) are short non-coding RNAs with therapeutic possibilities in cancer. The current study compared levels of 1510 miRNAs in prostate cancer bone metastases with corresponding levels in the normal prostate and in localized prostate tumors, and found downregulation of a set of miRNAs during disease progression (miRNA-1, -23c, -143-3p, -143-5p, -145-3p, -205-5p, -221-3p, -222-3p and -4328). The downregulation of those miRNAs during the development of bone metastases may lead to a loss of inhibitory effects on prostate cancer cells. Functional studies confirmed the inhibiting effects of miRNA-23c and -4328 on prostate cancer cell growth in culture, while no clear effects of miRNA-23c were observed on tumor growth in mice.

**Abstract:**

MicroRNAs (miRNAs) are aberrantly expressed in prostate cancer (PC), but comprehensive knowledge about their levels and function in metastatic PC is lacking. Here, we explored the differential expression of miRNA profiles during PC progression to bone metastasis, and further focused on the downregulation of miRNA-23c and -4328 and their impact on PC growth in experimental models. Using microarray screening, the levels of 1510 miRNAs were compared between bone metastases (*n* = 14), localized PC (*n* = 7) and benign prostate tissue (*n* = 7). Differentially expressed miRNAs (*n* = 4 increased and *n* = 75 decreased, *p* < 0.05) were identified, of which miRNA-1, -23c, -143-3p, -143-5p, -145-3p, -205-5p, -221-3p, -222-3p and -4328 showed consistent downregulation during disease progression (benign > localized PC > bone metastases). The downregulation of miRNA-23c and -4328 was confirmed by reverse transcription and quantitative polymerase chain reaction analysis of 67 metastasis, 12 localized PC and 12 benign prostate tissue samples. The stable overexpression of miRNA-23c and -4328 in the 22Rv1 and PC-3 cell lines resulted in reduced PC cell growth in vitro, and in the secretion of high levels of miRNA-23c (but not -4328) in extracellular vesicles. However, no tumor suppressive effects were observed from miRNA-23c overexpression in PC-3 cells subcutaneously grown in mice. In conclusion, bone metastases display a profound reduction of miRNA levels compared to localized PC and benign disease. The downregulation of those miRNAs, including miRNA-23c and -4328, may lead to a loss of tumor suppressive effects and provide biomarker and therapeutic possibilities that deserve to be further explored.

## 1. Introduction

Prostate cancer (PC) is a common disease with a complex etiology and a variable outcome. If detected early, patients can be cured by surgery or radiation. However, once the cancer metastasizes to other organs (mainly to the bone), curative treatment is no longer possible, although several palliative treatment options are available [1]. Deprivation of circulating androgens, which in turn induces growth arrest and apoptosis in tumor cells, is the first line of therapy for patients with metastatic PC. When the malignancy progresses, castration-resistant PC (CRPC) develops [2] that with limited success can be treated with chemotherapy and inhibitors of androgen receptor (AR) signaling [3]. To improve survival for patients with metastatic disease, novel therapeutic strategies are needed.

MicroRNAs (miRNAs) are endogenous non-coding RNAs (~22 nucleotides long) that direct Argonaute proteins (e.g., Ago2) to complementary regions of messenger RNAs (mRNAs), which in turn inhibits the translation of the targeted mRNAs and reduces their stability [4]. MiRNA levels are frequently up- or down-regulated in cancer and such alterations may be either oncogenic or tumor suppressive, by directly or indirectly affecting proteins that in turn regulate cancer hallmarks [5,6,7]. Furthermore, miRNAs are secreted in extracellular vesicles (EVs), by which they mediate cellular crosstalk between cancer cells and their microenvironment [8].

Tumor suppressive miRNAs may have therapeutic possibilities, and some miRNA-based therapies have already entered clinical trials [9,10,11,12,13]. An miRNA-34 mimic was proposed as a treatment agent for solid malignancies, but due to immune-mediated complications the trial was terminated [11]. In contrast, an miRNA-16 mimic with anti-tumor activity was better tolerated by pleural mesothelioma patients [10]. A similar strategy might be possible for treating metastatic PC, despite the aggressive nature of the disease. Many miRNAs have been reported as downregulated in PC compared to normal prostate tissue, and restorations of specific miRNAs have been shown to reduce PC cell growth in vitro [14,15,16,17,18,19,20,21,22].

Comprehensive knowledge of miRNA levels and functions in metastatic PC is lacking. Here, the current study aimed to identify miRNAs showing diverse expression patterns in normal prostate tissue during PC development and progression to bone metastatic disease. Subsequently, we focused on two miRNAs (-23c, -4328) to verify their downregulation in PC bone metastases and to explore their impact on PC growth in vitro and in vivo. Encouraged by a recent publication [23], we also specifically examined levels of miRNA-23c in EVs as well as the suggested inhibiting effect of miRNA-23c on tumor angiogenesis.

## 2. Materials and Methods

### 2.1. Patients

The study included 13 patients with localized PC who were treated with radical prostatectomy at Umeå University Hospital between 2008 and 2009, and 67 separate patients who were operated on for spinal cord compression due to metastatic PC at the Umeå University Hospital (2003–2014). The clinical characteristics of the metastatic patients are given in Table 1. At the time of the metastasis surgery and tissue collection, 15 of the patients were hormone-naïve (HN), 4 had been treated with androgen-deprivation therapy (ADT) for 1–3 days (short-term ADT), and 49 were defined as having CRPC. For the patients treated with radical prostatectomy, the mean age was 60 years (range 48–68 years) and the mean serum level of prostate specific antigen (PSA) was 11 µg/L (range 3.5–26 µg/L). Clinical local stage was T2 (*n* = 4) or T3 (*n* = 9) and Gleason score (GS) was 7 (*n* = 11) or 8 (*n* = 2). The patient cohorts have been previously described [24,25]. Patients gave their informed consent, and the study was conducted in accordance with the Declaration of Helsinki. The study was approved by the local ethics review board of Umeå University (Dnr 03-482, Dnr 03-158 with amendments Dnr 04-26M, Dnr 2013-57-31M).

### 2.2. Clinical Samples

The bone metastasis tissue samples were fresh-frozen in liquid nitrogen. Fresh radical prostatectomy specimens were received at the pathology department immediately after surgery and cut into 0.5 cm thick slices. From these slices, 20 samples were collected per patient using a 0.5 cm skin punch and frozen in liquid nitrogen within 30 min after surgery. The prostate slices were formalin-fixed, embedded in paraffin, cut in 5 µm thick sections, whole-mounted and stained with hematoxylin and eosin (H&E). Tissue sample composition (non-malignant or malignant) of the frozen pieces was determined from their location in the whole-mount sections. Tissue composition was confirmed by the microscopic examination of H&E-stained sections and then cryo-sectioned into extraction tubes.

### 2.3. Cell Lines

All cell lines were acquired from the American Type Culture Collection (ATCC, Manassas, VA, US) and maintained according to the manufacturer’s description in a humidified environment at 37 °C and 5% CO_2_. The 22Rv1, PC-3 and LNCaP cell lines were acquired to represent AR positive (22Rv1, LNCaP) and AR negative (PC-3) PC cells, respectively. The RWPE-1 and PNT1A cells were acquired to represent normal epithelial cells and WPMY-1 to represent stromal myoepithelial cells. All cell lines except RWPE-1 were cultured in RPMI-1640 + GlutaMAX medium supplemented with 10% heat-inactivated fetal bovine serum (FBS-HI), 10 mM HEPES, 1 mM of sodium pyruvate, 100 U penicillin/mL, and 100 µg streptomycin/mL (Gibco, Thermo Fisher Scientific, Waltham, MA, USA). RWPE-1 cells were cultured in keratinocyte serum-free medium with the recommended supplements (Gibco, Thermo Fisher Scientific). All cells were tested for mycoplasma infection, and the 22Rv1 and PC-3 cell lines were further tested for virus contaminations (IDEXX BioAnalytics, Westbrook, ME, USA).

### 2.4. Isolation of microRNA

The AllPrep DNA/RNA/Protein Mini Kit (Qiagen, Hilden, Germany) was used to isolate nucleic acids and the RNeasy MinElute Cleanup Kit (Qiagen) to enrich for short RNAs.

### 2.5. Microarray Analysis

Microarray hybridization was performed at the TATAA Biocenter (Gothenburg, Sweden) on the 3D-Gene^®^ microarray platform (Toray Industries, Tokyo, Japan). Synthetic miRNAs (miRNA spike-in controls, Toray Industries) were added in predefined concentrations and the samples (88ng each) were labelled, hybridized and washed (Toray miRNA Labelling Kit, Toray Industries). The 3D-Gene^®^ Scanner 3000 (Toray Industries) detected the relative fluorescent intensities of miRNAs (*n* = 2018) based on the miRBase 19 database (www.mirbase.org, accessed on 12 March 2015). Background normalization was performed on the scanner by excluding potential outliers, i.e., spots without probes, demonstrating the lowest and highest background (mean ± 2 standard deviations). The probes were corrected for the mean fluorescent intensity (relative fluorescent units, RFU) of the remaining blank spots. Finally, the samples were adjusted for systematic variations by global mean normalization. Prior to differential expression analysis, normalized data were log_2_ transformed and median-centered. MiRNAs that were undetected in all samples were excluded, leaving a total of 1510 miRNAs. Differentially expressed miRNAs (*p* < 0.05) were identified in R (v4.0.5) using the limma (v3.46.0) package, and visualized by heatmaps based on unsupervised cluster analysis.

### 2.6. Reverse Transcription Quantitative Polymerase Chain Analysis (RT-qPCR)

The TaqMan miRNA Reverse Transcription Kit (Applied Biosystems, Foster City, CA, USA), the TaqMan Universal Mastermix II No UNG (Applied Biosystems) and the TaqMan Small RNA Assays Primer and Probe Sets (assay ID: 463068_mat for miRNA-23c, (AUCACAUUGCCAGUGAUUACCC; MIMAT0018000 and assay ID: 243596_mat for miRNA-4328, CCAGUUUUCCCAGGAUU; MIMAT0016926, Applied Biosystems) were used to quantify relative levels of mature miRNAs. In each reaction, the reverse transcription of 5–10 ng of RNA was carried out using the Veriti 96-Well Thermal Cycler (Applied Biosystems). Cycle threshold (CT) values were determined in triplicates by employing the 7900 HT Real-Time PCR System (Ambion, Austin, TX, USA). The miRNA-23c and -4328 levels were normalized to the geometric mean of small nucleolar RNAs: the small nucleolar RNAs C/D Box 48 (RNU48, assay ID: 001006), C/D Box 47 (U47, assay ID: 001223) and H/ACA Box 66 (RNU66, assay ID: 001002). Relative quantities were determined by the ΔΔCT method. Undetected readings were assigned a threshold value of 40 cycles.

### 2.7. Lentiviral Transduction

The 22Rv1 and PC-3 cells were transduced with shMIMIC human lentiviral vectors (Horizon Discovery, Cambridge, UK) to stably overexpress miRNA-23c or -4328. The SMARTvector Non-Targeting Control (NTC) was expressed to serve as a negative control. The vectors contained a turbo green fluorescent protein (turboGFP) and a puromycin resistance gene cassette. After transduction in 6-well plates, cells were cultured in a medium supplemented with 5 µg/mL puromycin (Takara Bio Inc., Tokyo, Japan) for antibiotic selection. Overexpression was confirmed by monitoring the turboGFP by fluorescence microscopy and by RT-qPCR analysis of miRNA-23c and -4328 levels, as described above.

### 2.8. Proteomic Profiling and Quantification

Relative protein quantification was performed to compare protein expression in 22Rv1 and PC-3 single cell clones overexpressing miRNA-23c and -4328, compared to corresponding NTC cells. Single cell clones overexpressing either miRNA-23c (*n* = 3), -4328 (*n* = 3) or NTC (*n* = 3) were analyzed in triplicate. Details of the sample analysis are described in the supplementary methods.

In short, samples were homogenized by bead-beating, proteins were extracted and equal amounts from samples and references (representative pools of 22Rv1 or PC-3 cell samples) were digested by trypsin. Samples were labeled using 10-plex tandem mass tagging (TMT) and combined into two sets. The sets were fractionated by basic reversed-phase chromatography, and analyzed with nano LC-MS/MS on an Orbitrap Fusion™ Tribrid™ mass spectrometer (Thermo Fisher Scientific). Identification and relative quantification were performed using Proteome Discoverer (v2.4, Thermo Fisher Scientific) against *Homo Sapiens* (Swiss-Prot Database, November 2020) using Mascot 2.5 (Matrix Science) as a search engine. For quantification, TMT reporter intensity values for each sample were normalized at the total peptide amount. The reference samples were used as the denominator and for the calculation of the ratios. Quantified proteins were filtered at a 5% false discovery rate (FDR) and exported for downstream analysis.

Briefly, normalized abundance ratios (Appendix A) were variance-stabilized and then used to carry out differential enrichment analysis on R (v4.0.5) using the DEP (v1.20.0) package [26]. Differentially expressed proteins were identified relative to the NTC, using the Benjamini–Hochberg method to adjust *p*-values for FDR (*adjp*), visualized by volcano plots, and compared with predicted theoretical miRNA targets according to the TargetScan database [27]. Suggested protein functions were elucidated by using the Universal Protein Resource Database (www.uniprot.org, accessed on 3 February 2022).

### 2.9. Gene Set Enrichment Analysis

Gene ranks were calculated by using *t*-statistic acquired from differential expression analysis results. The gene set enrichment analysis (GSEA) of the pre-ranked gene list was performed using the GSEA Software (v4.1.0). The enrichment of hallmark gene sets (*n* = 50, v7.4) acquired from the Molecular Signatures Database (MSigDB) [28] was quantified and presented as normalized enrichment scores (NES). An FDR of ≤0.25 was used to select significant gene sets.

### 2.10. Cell Growth Rate in Culture

The relative cell proliferation rate was determined using two different methods, as follows. (1) The fluorescence intensity of turboGFP was measured over 96 h in replicates (*n* = 12), using the Spectramax i3x microplate reader (Molecular Devices, San Jose, CA, USA) with the Minimax 300 imaging cytometer with a wavelength 456 for excitation and 541 for emission. The relative increase in turboGFP was compared to the intensity at the first measurement (at ~15–18 h). (2) Metabolically active cells were analyzed with the CellTiter-Glo™ Luminescent Cell Viability Assay (Promega Corp., Madison, WI, USA), according to the manufacturer’s instructions. Cells were cultured in replicates (*n* = 6), in opaque white 96-well plates, 1 × 10^4^ and 1.5 × 10^3^ cells/well for the 22Rv1 and PC-3 cell lines, respectively. After 72 h, the CellTiter-Glo™ reagent (Promega Corp.) was added and luminescence was determined on the SpectraMax i3x (Molecular Devices) microplate reader.

### 2.11. Cell Response to Enzalutamide and Simvastatin

The 22Rv1 cells overexpressing miRNA-23c, -4328 or NTC as well as the wildtype cell line were seeded in 96-well plates in different concentrations (*n* = 8) of Enzalutamide ranging from 0 to 1000 µM (Selleck chemicals, Houston, TX, USA) or Simvastatin ranging from 0 to 100 µM (Sigma Aldrich, Saint Louis, MO, USA). Cell viability was evaluated after 96 h with the CellTiter-Glo™ assay (Promega Corp.). Curve fitting with nonlinear regression and determination of the half maximal inhibitory concentration (IC_50_) was performed (GraphPad software v9, San Diego, CA, USA).

### 2.12. Wound Healing Assay

The 22Rv1 and PC-3 cells overexpressing miRNA-23c or -4328 and the corresponding NTC cells were seeded in 12-well plates in 8 × 10^5^ cells or 4 × 10^5^ cells per well, respectively. At 24 h after seeding, replicate scratches (n = 3–4) were made using a 100 µL pipette tip. The wells were washed 2 times with fresh medium to remove debris. The scratches were photographed under light microscopy at regular intervals up to 36 h. The scratch area was quantified using the Image J plugin for scratch wound healing assays [29]. The relative cell migration was calculated as the relative difference in scratch area compared to the NTC.

### 2.13. Isolation of Extracellular Vesicles

Briefly, 22Rv1 and PC-3 cells overexpressing miRNA-23c, -4328 or the NTC were seeded in T175 flasks in EV-depleted medium (2 × 10^7^ or 3 × 10^6^ cells/flask, for 22Rv1 and PC-3 cells, respectively). To obtain EV-depletion, FBS-HI (Gibco™, Thermo Fisher Scientific, Waltham, MA, USA) had been pre-centrifuged at 100,000× *g* using a swing bucket 32 titanium rotor (Beckman Coulter, Brea, CA, USA) for 18 h to pellet EVs of bovine origin. After 72 h, the conditioned medium was carefully removed and centrifuged at 300× *g* at room temperature for 20 min to pellet live cells, followed by 2000× *g* for 20 min at 4 °C to pellet apoptotic bodies and large debris, and 10,000× *g* for 30 min at 4 °C to remove large vesicles. The cleared cell conditioned media were stored at −80 °C until the timepoint for the isolation of EVs using size exclusion chromatography.

Before size exclusion chromatography, the cleared conditioned medium were concentrated with the Amicon^®^ Ultra-15 centrifugal filter devices (Merck Millipore, Burlington, MA, USA). EVs were then separated from proteins using the Izon original or Izon qEV2 columns (Izon Science, Christchurch, New Zealand), according to the manufacturer’s protocol. Protein concentration in the collected fractions was determined by the DeNovix DS-11 spectrophotometer (DeNovix Inc., Wilmington, DE, USA). Protein-free fractions were defined to contain vesicles according to the manufacturer’s description. Those were therefore pooled and used separately from the protein-rich fractions in downstream assays.

### 2.14. Analysis of Extracellular Vesicles

After size exclusion chromatography, vesicle size and the concentration of protein-free fractions were determined by nanoparticle tracking analysis using the NanoSight NS300 system (NTA software version 3.4, Malvern panalytical, Malvern, UK). Samples were diluted 1:20 to obtain a total of 20–100 particles per frame (laser setting of 488 nm, camera level of 12 and detection threshold of 5). Samples were injected with a syringe pump at a flow speed of 30. Three captures with a duration of 30 sec each were acquired for each sample diluted in triplicates, and for statistical calculations the average was used.

RNA was extracted from pooled, protein-free fractions using the qEV RNA extraction kit (Izon Science) and levels of miRNA-23c and -4328 were determined by RT-qPCR, as described above, with RNU48 and U47 serving as endogenous controls. Before RNA extraction, pooled fractions were concentrated using Amicon^®^ Ultra-15 centrifugal filter devices (Merck Millipore).

### 2.15. Animal Study

PC-3 cells (1 × 10^6^) overexpressing miRNA-23c or NTC were diluted 1:1 in RPMI-1640 + GlutaMAX medium (Thermo Fisher Scientific) and Matrigel (BD Biosciences, Franklin Lakes, NJ, USA) and injected subcutaneously into the flanks of 7-week-old, athymic, male BALB/c nude mice (*n* = 10 NTC and *n* = 9 miRNA-23c) (Charles River, Germany). Tumor widths (w) and lengths (l) were measured with a caliper two to three times per week, and tumor volumes (V) were calculated by using the following formula: V = l × (w^2^)/2. The experiment was terminated when the first tumor reached a volume of about 1000 mm^3^. Tumors were dissected and a small part of each tumor was fresh-frozen in liquid nitrogen, and the rest was fixed in 4% paraformaldehyde followed by paraffin embedment. Animal work was carried out in accordance with the protocol approved by the Umeå Ethical Committee for Animal Studies (permit number A16-2020).

### 2.16. Immunohistochemistry

Formalin-fixed paraffin-embedded tumor sections (4 µm) were deparaffinated according to standard procedure and stained with H&E for the evaluation of tumor histology. Parallel sections (6 µm) were immunostained for CD31/PECAM-1 (AF-3628, 4 µg/mL) using the automated Discovery ULTRA system with Cell Conditioning 1 (CC1) as antigen retrieval buffer, and detected using the OmniMap anti-goat-HRP with ChromoMap DAB Detection kit (Roche, Basel, Switzerland). Stained sections were scanned with the Pannoramic P250 FI scanner (3DHistech, Budapest, Hungary). Vessel density was quantified using a square lattice in the open source QuPath software (v0.4.2) [30] by counting the fraction of grid intersection falling on the stained tissue component. Three to nineteen areas (440,000 µm^2^, 0.44 mm^2^ each) were evaluated in relation to tumor size.

## 3. Results

### 3.1. Differential Expression of miRNAs in Metastatic Prostate Cancer

Microarray analysis was performed to determine the levels of 2018 miRNAs (miRBase 19 database, www.mirbase.org) in bone metastasis (*n* = 14), localized PC (*n* = 7) and benign prostate (*n* = 7) tissue samples. A total of 1510 miRNAs were detected (Appendix A). Using differential expression analysis, 79 miRNAs were identified as significantly up (*n* = 4) or downregulated (*n* = 75) in bone metastasis and localized PC tissue samples were compared to benign prostate tissue samples (*p* < 0.05, Appendix A). Nine of the seventy-nine differentially expressed miRNAs showed consistent downregulation during PC development and progression into metastatic disease according to all comparisons carried out (localized PC vs. benign; metastatic vs. localized PC; metastatic vs. benign (*p* < 0.05) (Figure 1a). Seven of these miRNAs had been previously reported as downregulated in PC (miRNA-1, -143-3p, -143-5p, -145-3p, -205-5p, -221-3p and -222-3p [14,16,18]), while no comparisons for miRNA-23c and -4328 were found in the literature. Therefore, miRNA-23c and -4328 were selected for further analysis.

The relative fluorescence intensities of miRNA-23c and -4328, as determined by microarray analysis, are shown in Figure 1b. The miRNAs were downregulated in localized PC as compared to benign tissue (*p* < 0.05) and in bone metastases as compared to localized PC (*p* < 0.0001) and benign tissue (*p* < 0.05) (Figure 1b). Using RT-qPCR analysis of an extended number of samples (bone metastases, *n* = 67; localized PC, *n* = 12; benign prostate tissue, *n* = 12), miRNA-23c and -4328 were confirmed to be downregulated in bone metastases (*p* < 0.05 and *p* < 0.0001, respectively) but not in localized PC (*p* > 0.05) (Figure 1c).

#### Metastatic Levels of miRNA-23c and -4328 in Relation to Clinicopathological Variables

Despite the general reduction in miRNA-23c and -4328 in bone metastatic PC, levels of both miRNAs showed a large variation between cases (Figure 1c). This variation was analyzed in relation to metastasis and patient characteristics, as shown in Table 1. No obvious relation was found between any of the miRNAs and patient age, serum PSA levels, primary tumor differentiation (Gleason score) or survival (Table 1). Levels of miRNA-23c and -4328 were significantly higher in CRPC than in HN metastases (*p* < 0.05), and miRNA-4328 also showed increased levels in short-term ADT-treated cases (*p* < 0.05) (Table 1).


Figure 1Reduced miRNA levels during prostate cancer disease progression. (**a**) Heatmap displaying unsupervised hierarchical clustering of biopsies (*n* = 28) and miRNAs (*n* = 9) with significantly reduced tissue levels in localized prostate cancer (PC) (*n* = 7) as compared to adjacent benign prostate tissue (*n* = 7), and in castration-resistant prostate cancer bone metastases (bone mets) (*n* = 14) as compared to localized PC or benign prostate tissue (*p* < 0.05, differential expression analysis, in R (v4.0.5) limma (v3.46.0) package). (**b**) Relative levels of miRNA-23c and -4328, respectively, based on normalized fluorescent intensities in microarray analysis (numbers and statistics as indicated in (**a**)). (**c**) Relative levels of miRNA-23c and -4328, respectively, based on reverse transcription and quantitative polymerase chain analysis of benign prostate tissue (*n* = 12), localized PC (*n* = 12) and bone metastases (*n* = 67), including hormone-naïve (*n* = 15), short-term treated (*n* = 4) and CRPC (*n* = 48) cases (see Table 1). * *p* < 0.05, ** *p* < 0.01, *** *p* < 0.001, **** *p* < 0.0001, ns *p* > 0.05 (Mann–Whitney U-test and Wilcoxon test for paired observations). The square and triangles indicate outliers in the Tukey box plots. X values represent extreme outliers outside the scale.
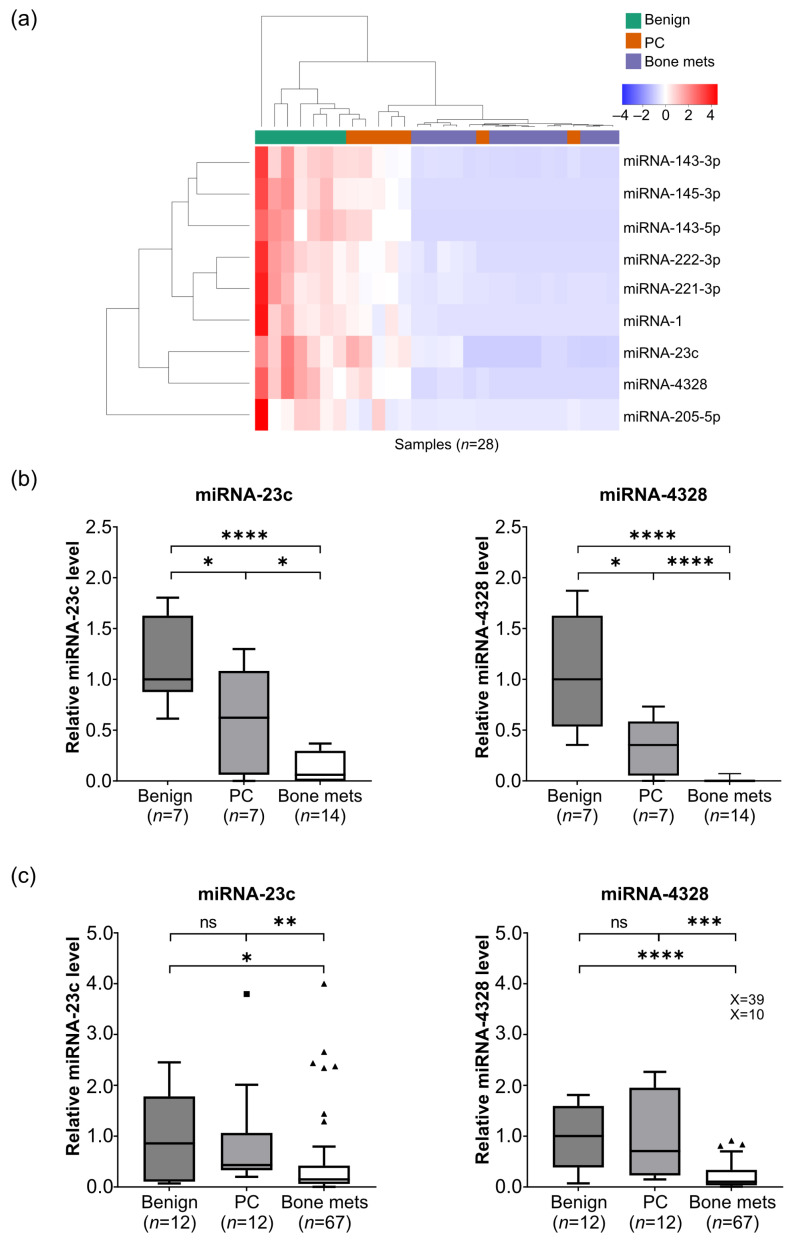



### 3.2. Forced Overexpression of miRNA-23c and -4328 in Prostate Cancer Cell Lines

Wildtype 22Rv1 and PC-3 cells showed low expression levels of miRNA-23c and -4328 in comparison to prostate cell lines representing normal epithelial cells (RWPE-1, PNT1A), myoepithelial cells (WPMY-1) and lymph metastasis derived PC cells (LNCaP) (Appendix A), and were selected as model systems for aggressive PC to be used for subsequent functional studies. Lentiviral transduction followed by antibiotic selection was carried out in 22Rv1 and PC-3 cell lines to induce the stable overexpression of miRNAs-23c or -4328, with the hypothesis that this would make the cells less aggressive. RT-qPCR analysis confirmed highly increased levels of miRNA-23c in 22Rv1 (400-fold, *p* < 0.0001) and PC-3 (500-fold, *p* < 0.01) cells. A corresponding increase in miRNA-4328 was 100-fold in 22Rv1 cells (*p* < 0.01) and 140-fold in PC-3 cells (*p* < 0.01) compared to NTC and wildtype cells (Figure 2a,b).

#### 3.2.1. Proteomic Effects of miRNA-23c and -4328 Overexpression

Proteomic analysis was performed to compare protein profiles of 22Rv1 and PC-3 cells following miRNA-23c or -4328 overexpression with profiles of the corresponding NTC cells. A total of 5142 (22Rv1) and 4814 (PC-3) proteins, respectively, were identified and quantified using LC-MS/MS (Appendix A). In the 22Rv1 cells, miRNA-23c overexpression yielded a significant downregulation of nineteen proteins (*adjp* < 0.05), of which five (A1BG, ERLIN2, RPIA, WBP2, YOD1) were predicted as theoretical targets in the TargetScan database (Appendix A). Moreover, 18 proteins were significantly upregulated by miRNA-23c overexpression in 22Rv1 cells (Appendix A). MiRNA-4328 overexpression in 22Rv1 cells generated up- (*n* = 4) and downregulated (*n* = 9) proteins, with the majority (*n* = 8) predicted as targets (ADAM9, ATG2A, BARD1, FMR1, ITIH4, MFN1, PIKFYVE, TRABD) (Appendix A). The corresponding data for PC-3 cells were up- (*n* = 10) and downregulated (*n* = 21) proteins including five anticipated targets (APAF1, LMNB1, MED14, PJA1, STIM2) after miRNA-23c overexpression (Appendix A) and up- (*n* = 5) and downregulated (*n* = 28) proteins with six anticipated targets (UNC80, FERMT2, ART4, DIMT1, ITIH4, PSMA4) after miRNA-4328 overexpression (Appendix A). Notable, only one protein (ITIH4) was significantly downregulated in both cell lines following the overexpression of miRNA-4328, and only two proteins (AHSG and APOH) were significantly downregulated in both cell lines following the overexpression of miRNA-23c. Suggested functions of the proteins being theoretical targets for miRNAs-23c or -4328 and which showed significant downregulation (*p* < 0.05) after overexpression of the corresponding miRNA are listed in Appendix A, as direct citations from the Universal Protein Resource Database (www.uniprot.org, accessed on 3 February 2022).

#### 3.2.2. Functional Effects of miRNA-23c and -4328 Overexpression

To identify possible functional effects of miRNA-23c and -4328 overexpression in 22Rv1 and PC-3 cells, protein profiles from the LC-MS/MS analysis were used to calculate enrichment scores of hallmark gene sets (*n* = 50, v7.4, MSigDB). The overexpression of miRNA-23c resulted in three positively and five negatively enriched hallmarks in 22Rv1 cells (*adjp* < 0.25) (Appendix A) and two positively and eleven negatively enriched hallmarks in PC-3 cells (*adjp* < 0.25) (Appendix A). The negative enrichment of “glycolysis” and “coagulation” were consistently seen in both 22Rv1 and PC-3 cells following miRNA-23c overexpression (Appendix A). The overexpression of miRNA-4328 displayed thirteen positively and ten negatively enriched hallmarks in 22Rv1 cells (*adjp* < 0.25) (Appendix A) and one positively and four negatively enriched hallmarks in PC-3 cells (*adjp* < 0.25) (Appendix A). The positive enrichment of “early estrogen response” and the negative enrichment of “KRAS signaling”, “glycolysis” and “complement” were consistently seen in both cell lines following miRNA-4328 overexpression (Appendix A).

Since many of the deregulated hallmarks showed the potential for affecting cell growth and viability, those processes were evaluated in cell culture after the overexpression of miRNA-23c and -4328, respectively, as delineated below. The positive enrichment of “androgen response” and “cholesterol homeostasis” in the 22Rv1 cell line after miRNA-4328 overexpression motivated us to explore its response to Enzalutamide (an AR antagonist) and Simvastatin (an inhibitor of cholesterol synthesis). However, no difference in the cell response to Enzalutamide or Simvastatin could be seen after either miRNA-23c or -4328 overexpression by evaluating the IC_50_ (Appendix A). Furthermore, the negative enrichment of “epithelial to mesenchymal transition” in PC-3 cells following miRNA-23c overexpression and in 22Rv1 cells following miRNA-4328 overexpression motivated us to perform a wound healing assay to evaluate the possible effects of the miRNAs on cell migration. As can be seen in Appendix A, however, the overexpression of miRNA-23c or miRNA-4328 did not significantly affect the migration of PC-3 cells, and the 22Rv1 cells did not migrate in this assay.

#### 3.2.3. Overexpression of miRNA-23c and -4328 Reduces Prostate Cancer Cell Growth in Culture

The growth rates of 22Rv1 and PC-3 cells stably overexpressing miRNA-23c or -4328 were compared to the growth rate of NTC cells using two different methods. When cell growth was measured by the live cell imaging of fluorescence intensity over time (turboGFP), cells overexpressing miRNA-23c or -4328 showed a slower proliferation rate than their corresponding NTC cells (17–30% at 96 h, *p* < 0.0001) (Figure 2c). Accordingly, reduced cell growth after miRNA-23c or -4328 overexpression was seen when the ATP production was measured using CellTiter-Glo^®^ luminescent cell viability assay as an indicator of the number of viable cells (8–14% *p* < 0.0001 (22Rv1) and 14–17% *p* < 0.001 (PC-3) at 72 h) (Figure 2d).

### 3.3. Extracellular Vesicles Are Highly Enriched for miRNA-23c

To explore the possibility that miRNA-23c or -4328 could reach the tumor microenvironment by secretion via EVs, the levels of miRNA-23c and -4328 were determined using RT-qPCR analysis of vesicle containing fractions (fractions 2–10, mode vesicle size 88.5–134 nm), which were separated from protein-rich fractions (fractions ≥11) by size-exclusion chromatography of the conditioned cell media (Appendix A). The relative level of miRNA-23c in EVs isolated from overexpressing 22Rv1 and PC-3 cells was about 150 and 2000 times higher (*p* < 0.0001), respectively, than in EVs isolated from the corresponding NTC cells (Figure 3). The miRNA-4328 showed no increase in EVs isolated from overproducing 22Rv1 cells, and only a moderate (2.8-fold, *p* < 0.001) increase in EVs isolated from overproducing PC-3 cells (Figure 3).

### 3.4. No Reduction in Tumor Cell Growth in Mice by miRNA-23c Overexpression

The high miRNA-23c levels in EVs secreted from miRNA overproducing tumor cells encouraged us to explore the possible effects of miRNA-23c overexpression on tumor growth and angiogenesis in vivo. For this purpose, PC-3 cells overproducing miRNA-23c were subcutaneously inoculated into nude mice. Tumor growth was monitored over time and compared to the growth rate of NTC cells. No difference in tumor take, growth rate, or volume at endpoint was observed (Figure 4a,b). In addition, no obvious difference in tumor histology or vessel density was observed between the miRNA-23c overproducing and the NTC tumors (Figure 4c,e). RT-qPCR analysis confirmed the overexpression of miRNA-23c (on average, 75-fold compared to NTC, *p* < 0.0001) in established PC-3 tumors (Figure 4d).

## 4. Discussion

The current study has identified miRNA profiles downregulated in metastatic PC, of which some may possess tumor suppressive activities and have the potential to be developed into novel therapeutic strategies. The microarray screening identified 79 miRNAs, dysregulated during malignant transformation and disease progression, and the majority (95%) showed reduced levels in PC bone metastases and localized disease as compared to benign tissue. This prominent downregulation of miRNAs in PC bone metastases was in line with results previously reported for PC [14,15,16,18,31], and might be the result of a hypoxic bone environment as hypoxia has been shown to globally reduce miRNA expression due to the downregulation and/or inactivation of Drosha, Dicer and Ago2 (reviewed in [7]). Of the four miRNAs (miRNA-345-3p, -4506, -4525 and -5006-5p) that were found to be consistently upregulated in localized and metastatic PC compared to benign prostate tissue, miRNA-345-3p was already known to be upregulated in PC [14]. Levels of miRNA-4506, -4525 and -5006-5p had not been reported previously in relation to PC, and remain to be verified by further studies.

Of the 75 suppressed miRNAs, a limited set consisting of miRNA-1, -23c, -143-3p, -143-5p, -145-3p, -205-5p, -221-3p, -222-3p and -4328 were considered of specific interest as they demonstrated a stepwise downregulation from benign prostate to adjacent PC tissue and, furthermore, from localized PC to bone metastases. All but two (miRNA-23c and -4328) of those nine miRNAs had been previously reported as downregulated in PC, indicating the accuracy of our microarray screening study [14,16,18].

MiRNA-1-3p is transcribed in the same cluster as miRNA-133a and has previously been described as epigenetically silenced in PC and further downregulated in metastasis [32]. Additionally, miRNA-1-3p has been shown to function as a tumor suppressor, confirmed in vitro and in vivo [32,33]. Tumor suppressive roles of the miRNA-143/145 cluster have also been previously confirmed [34]. Furthermore, the well-studied miRNA-145 and -205-5p have been demonstrated to inhibit the proliferation of PC cells in culture [21,35,36,37,38], and the miRNA-221/222 cluster to affect the migration and invasion of PC cell lines in vitro and to be downregulated in PC metastasis samples [39].

The seed sequence of miRNA-23c is shared with miRNA-23a-3p and miRNA-23b-3p, and all three by definition belong to the same miRNA family [40]. MiRNA-23a-3p and -23b-3p are transcribed from different genomic clusters (www.mirbase.org, accessed on 1 February 2023). The latter has been suggested as a tumor suppressor, inhibiting PC cell migration, invasion and metastasis [41,42]. Motivated by those results, we set out to verify the downregulation of miRNA-23c and -4328 during PC disease progression and to examine the functional effects of overexpressing miR-23c and -4328 in PC cells.

The downregulation of miRNAs-23c and -4328 in metastatic PC was verified by the RT-qPCR analysis of an extended set of samples, including not only bone metastatic samples from CRPC patients, but also samples from non-castrated HN patients and patients treated for a few days with castration. In contrast, we could not confirm any downregulation from benign tissue to localized PC, indicating that the microarray screen would have benefitted from including a larger set of patient samples. Additionally, it would have been preferable to include primary tumor and metastasis samples from the same patient, to increase the power of identifying consistently up- or downregulated miRNAs by taking individual variability into account, but unfortunately no such sample pairs were available. Notably, the increased levels of miRNA-23c and -4328 in CRPC compared to HN bone metastases (and for miRNA-4328 the increased levels in short-term castrated patients) indicate a possible regulation of those miRNAs by androgens.

The forced expression of miRNA-23c and -4328 in the 22Rv1 and PC-3 cell lines resulted in reduced protein levels of many theoretical targets for each miRNA, but also in deregulated levels of other proteins that could either be indirect effects of the directly altered miRNA profiles or non-target effects. The reason behind the upregulation of a few theoretical targets for miRNA-23c and -4328 is not known. The introduction of high miRNA levels might give off-target effects such as the binding of miRNA to unspecific mRNAs, the saturation of the RNA interference machinery, or immune responses, and all these undesired effects could possibly repress cell growth [43,44].

The GSEA of the proteomic profiles indicated alterations in cell metabolism and cell growth following miRNA-23c or -4328 overexpression, which were experimentally confirmed as reduced cell growth in both cell lines in culture. The 22Rv1 cells are androgen sensitive, but not androgen dependent due to the expression of AR and constitutively active AR variants [45]. The PC-3 cell line lacked the AR and luminal differentiation markers [46]. The growth reduction induced by both miRNAs in such diverse cell lines indicated intrinsic effects, not critically dependent on other main growth regulators such as the AR status. Results from the GSEA also motivated us to examine whether miRNA-4328 overexpression in 22Rv1 cells specifically affected the cellular response to Enzalutamide or Simvastatin, as this would have indicated a potential for miRNA-4328 to have an impact on conventional therapies targeting the AR. Unfortunately, no effects on androgen response or cholesterol homeostasis could be confirmed from comparing the IC_50_ values for Enzalutamide and Simvastatin in cells with and without miRNA-4328 overexpression. Assessing the cholesterol and androgen levels following miRNA overexpression in 22Rv1 cells might instead have been informative. Notably, the 22Rv1 cells showed a limited response to Enzalutamide due to the expression of constitutively active AR variants [47].

To be able to evaluate not only the direct effects of miRNA-23c overexpression on PC cell growth, but also possible indirect effects mediated through the microenvironment, PC-3 cells were grown subcutaneously in male mice. We hypothesized that miRNA-23c overexpression would inhibit the tumor growth of PC-3 in vivo, partly due to the growth inhibiting effects seen in vitro, but also due to an anticipated inhibitory effect of miRNA-23c on angiogenesis, as previously reported [23]. Additionally, the enrichment of miRNA-23c in EVs, and the lack of a corresponding increment of miRNA-4328, suggested a likelihood of miRNA-23c affecting not only the cells of origin, but also cells within the microenvironment and processes such as angiogenesis and metastasis [48,49]. Thus, we were disappointed to find that miRNA-23c overexpression did not inhibit PC-3 tumor growth in mice, nor affected the tumor morphology or the tumor vessel density.

Limitations of the study include a poor specificity of the microarray analysis in differentiating miRNAs with similar sequences [50]. RNA sequencing analysis might thus have been a better choice of method for the screening study part. This would not only have given better resolution between miRNAs with similar sequences, but also the possibility for deep annotation of miRNA-23c and -4328 in prostatic tissue and an evaluation of their accuracies as miRNAs. According to published data, they both lack reads from the second 5′ arm of the hairpin precursor, a requisite for the annotation of miRNAs according to more strict criteria [51,52]. The interpretation of those caveats is not clear. Nevertheless, the suggested miRNAs-23c and -4328 have been scientifically studied and suggested to be of functional relevance in cancer [23,53,54,55,56,57,58,59]. The tissue localization of the miRNA-23c and -4328 will need to be confirmed. So far, this has proven difficult in our hands using in situ hybridization, but RT-qPCR analysis indicated the expression of miRNA-23c and -4328 in cells of both epithelial and stromal origin.

## 5. Conclusions

The miRNA-23c and -4328 are two among many downregulated miRNAs observed in clinical PC bone metastasis when compared to localized PC and benign prostate. The downregulation of those miRNAs may result in the loss of tumor suppressive effects on PC cells and provide biomarker and therapeutic possibilities that deserve to be further explored. The overexpression of miRNA-23c and -4328 moderately inhibited the cell growth and metabolism of 22Rv1 and PC-3 cells in vitro, but miRNA-23c overexpression had no clear effect on PC-3 tumor growth in vivo. The miRNA-23c, but not miRNA-4328, was secreted in high levels in Evs, suggesting a selective incorporation mechanism for miRNA-23c into Evs. Functional studies of Evs enriched for miRNA-23c are needed to evaluate the role of miRNA-23c in mediating cell to cell, as well as systemic effects. Additionally, future studies should examine the impact of miRNA-23c and miRNA-4328 on PC metastasis to the bone. Such experiments may involve not only upregulation, but also the downmodulation of miRNA-23c and -4328 in different model systems for metastatic PC.

## Figures and Tables

**Figure 2 cancers-15-02437-f002:**
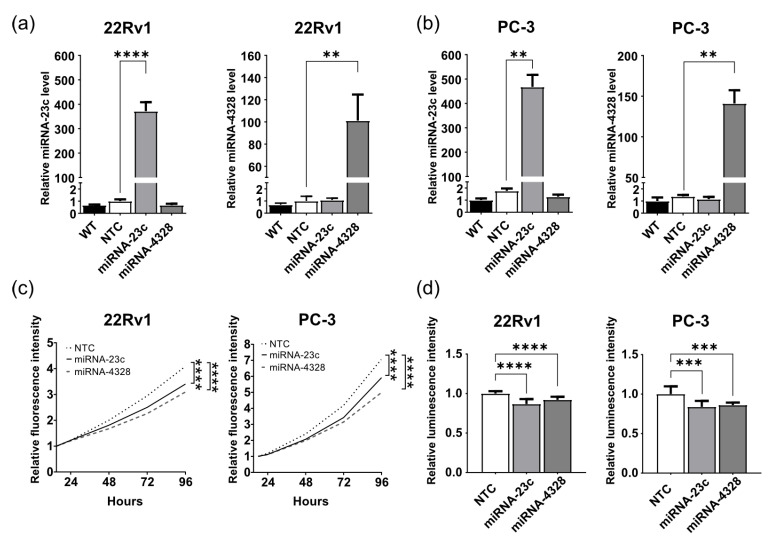
Overexpression of miRNA-23c and miRNA-4328 results in reduced prostate cancer cell growth in culture. (**a**,**b**) Relative levels of miRNA-23c and miRNA-4328 (mean ± standard deviation, *n* = 2–3) determined by reverse transcription and quantitative polymerase chain analysis after stable overexpression in the 22Rv1 and PC-3 cell lines. WT, wildtype; NTC, non-target control. (**c**) Relative proliferation rate of 22Rv1 cells and PC-3 cells overexpressing miRNA-23c, miRNA-4328 and NTC sequence, determined by measuring the fluorescent intensity of turbo green fluorescent protein (Spectramax i3x reader) at 24, 48, 72 and 96 h after seeding in relation to the intensity at 15 and 18 h, respectively (*n* = 12). (**d**) Relative number of 22Rv1 and PC-3 cells at 72 h after seeding (CellTiter-Glo^®^ Luminescent Cell Viability Assay). The experiment was repeated twice with six replicates each per cell line. ** *p* < 0.01, *** *p* < 0.001, **** *p* < 0.0001 (unpaired *t*-test).

**Figure 3 cancers-15-02437-f003:**
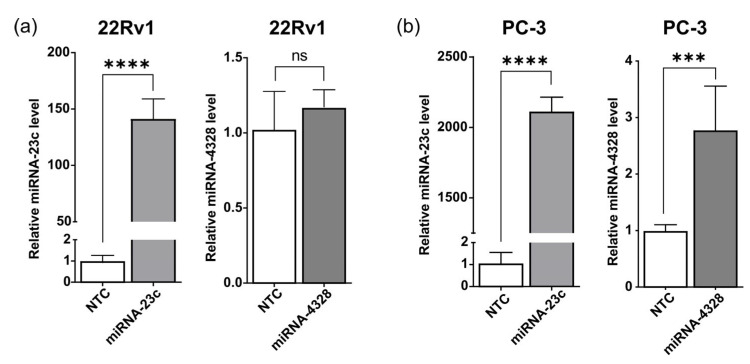
Relative levels of miRNA-23c and -4328 in extracellular vesicles from (**a**) 22Rv1 and (**b**) PC-3 cells overexpressing miRNA-23c or -4328, isolated using size exclusion chromatography (Izon qEV columns). Levels of miRNA were determined with reverse transcription and quantitative polymerase chain analysis and expressed in relation to levels in the non-target control (NTC) cells. *** *p* < 0.001, **** *p* < 0.0001, ns *p* > 0.05 (unpaired *t*-test).

**Figure 4 cancers-15-02437-f004:**
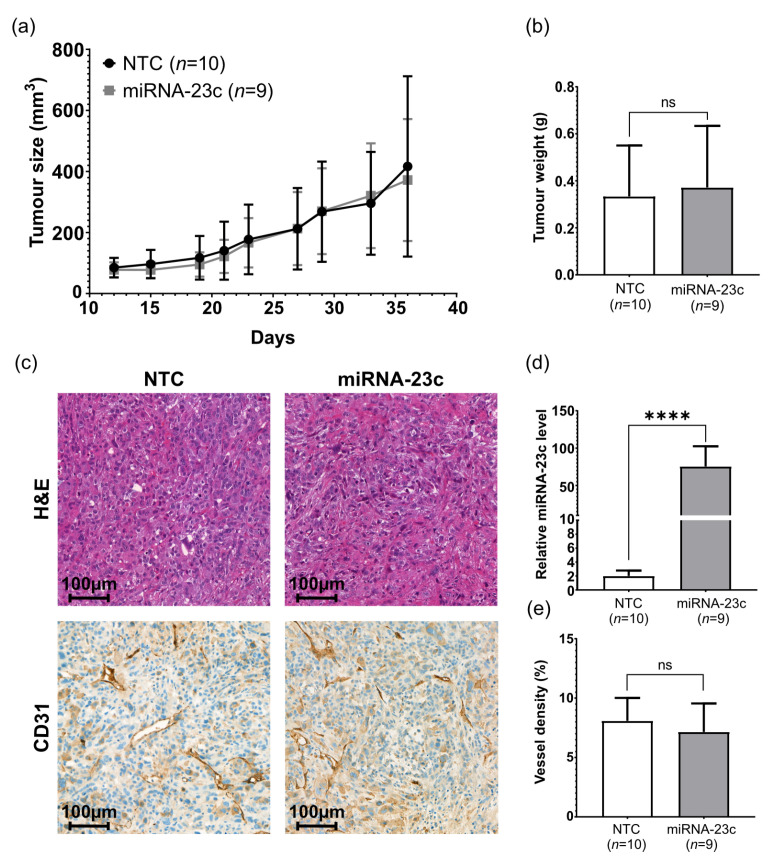
Effect of miRNA-23c overexpression on PC-3 tumor growth and histology in vivo. (**a**) Tumor size after subcutaneous injection of miRNA-23c overexpressing PC-3 cells and non-target control cells (NTC) in nude mice. (**b**) Tumor weight at endpoint (day 37). (**c**) Tumor histology at endpoint, as demonstrated by representative tissue sections from miRNA-23c overexpressing and NTC PC-3 tumors stained by hematoxylin and eosin (H&E) or CD31, as indicated. (**d**) Relative expression level of miRNA-23c in tumors at endpoint and (**e**) vessel density (%) in NTC and miRNA-23c overexpressing tumors at endpoint, according to volume density of CD31 stained structures. Data are shown as mean ± SD. **** *p* < 0.0001, ns *p* > 0.05 (unpaired *t*-test and Mann–Whitney U-test).

**Table 1 cancers-15-02437-t001:** Clinical characteristics of patients with bone metastatic prostate cancer in relation to relative levels of miRNA-23c and -4328.

Characteristics		miRNA-23c (*n* = 67)	miRNA-4328 (*n* = 67)
Age diagnosis (yrs)	69 (63; 76)	0.17 (0.072; 0.48)	0.10 (0.034; 0.30)
Age metastasis surgery (yrs)	73 (67; 79)	-	-
Serum PSA diagnosis (µg/L)	110 (47; 750)	-	-
Serum PSA metastasis surgery (µg/L)	290 (86; 980)	-	-
Follow-up after androgen-deprivation therapy (months)	43 (25; 73)	-	-
Follow-up after metastasis surgery (months)	10 (3.0; 31)	-	-
Gleason score at diagnosis:			
7	18 (27%)	0.17 (0.061; 0.37)	0.086 (0.032; 0.34)
8–10	28 (42%)	0.19 (0.078; 0.85)	0.17 (0.044; 0.35)
Not available	21 (31%)	NA	NA
Castration therapy ^a^:			
None (hormone-naïve)	15 (22%)	0.13 (0.060; 0.17)	0.032 (0.025; 0.052)
Short-term ^b^	4 (6.0%)	0.42 (0.24: 0.46)	0.27 (0.13; 5.1) *
CRPC	48 (72%)	0.21 (0.075; 0.71) *	0.13 (0.044; 0.31) *

Continuous variables given as median (25th; 75th percentiles). Categorical variables are given as numbers (%). Relative miRNA levels were assessed by reverse transcription and quantitative polymerase chain reaction and expressed in relation to levels in benign prostate tissue, as described in Materials and Methods. * *p* < 0.05 when compared to hormone-naïve (Mann–Whitney U-test). ^a^ Castration therapies given prior to collection of metastasis tissue samples included surgical ablation, LHRH/GnRH agonist therapy or bicalutamide treatment. ^b^ Castration therapy for 1 day to 3 days before metastasis tissue sampling. NA, not analyzed.

## Data Availability

Publicly available data sets were generated and analyzed in this study. The microarray data can be found here: [https://www.ncbi.nlm.nih.gov/geo/query/acc.cgi?acc=GSE230278 accessed on 16 April 2023]. The proteomic data can be found here: [http://www.ebi.ac.uk/pride/archive/projects/PXD041683 accessed on 16 April 2023].

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
