# Peer review of "Investigating microRNA Profiles in Prostate Cancer Bone Metastases and Functional Effects of microRNA-23c and microRNA-4328"

_cancers, 2023, doi:10.3390/cancers15092437_

Round 1

Reviewer 1 Report

This is a very well conducted and informative study. I have some concerns and comments – these are listed below.

1.       I suggest rephrasing the sentence on line 47, perhaps say ‘… is a common disease with a complex etiology’. Or perhaps use the terminology ‘heterogeneous’.

2.       I think it would have been more appropriate and meaningful for the authors to have used Abiraterone as well as Simvastatin to help determine whether these miRNA play a role in driving CRPC through mediating changes in cholesterol homeostasis – the former is a standard of care treatment used to treat CRPC and inhibits an enzyme, Cyp17, which plays a key role in testosterone synthesis. Did the authors assess cholesterol levels (and ideally also testosterone levels) following simvastatin treatment? 22Rv1 cells are known to synthesize testosterone via intracrine androgen synthesis, a key mechanism which drives CRPC. Assessing cholesterol and testosterone levels would be more informative than simply assessing the impact on cell proliferation. Please comment on this in the manuscript. The authors may also want to consider looking to see if there were any changes in protein expression levels of the enzymes involved in the testosterone synthesis pathway via analyses of their proteomics data.

3.       22Rv1 are known to be resistant to enzalutamide (https://www.ncbi.nlm.nih.gov/pmc/articles/PMC3549016/) due to expression of an AR splice variant. This could help explain why no response was observed. Please comment on this in the manuscript. For future studies, the authors should consider using a cell line that is responsive to Enzalutamide and/or using RNAi technology to knockdown all AR variants in the 22Rv1 cells.

4.       Did the proteomics analysis find any differences in expression of AR or PSA following forced overexpression of these miRNA plus minus treatment with enzalutamide? Assessing this would be a better way to follow up re. the functional effect of the observed enrichment for ‘androgen response’ genes rather than simply looking at the effect on cell proliferation. Please comment.

Reviewer 2 Report

Comments to the Author

Järemo and Wikström et al. have written a manuscript about the importance of miRNA in prostate cancer bone metastases. This manuscript identified a set of miRNAs were downregulated during progression from PC clinical samples via microarray assay. And the authors tested the function of miRNA-23c in vitro and in vivo. However, after reading this article there are still some problems needed to be addressed. Below are some detailed comments. 

Major:

1.     What are the mechanisms by which miRNA-23c and miRNA-4328 are downregulated during disease progression?

2.     Since the author focuses on the differential miRNAs in prostate cancer bone metastasis, the author needs to further validate the effects of miRNA-23c and miRNA-4328 on PC cell migration and invasion through experiments in vitro and animal experiments in vivo.

Minor:

1.     In Figure 2, a, b, c, d tag error. Missing d. 

2.     In figure 4c, the scale bar is not clear.

Reviewer 3 Report

In this manuscript by Järemo and Collaborators, a miRNA screen in healthy, PC and bone metastasis tissues was performed to evaluate possible roles of miRNAs in PC progression. Two miRNAs, miR-23c and miR4328, have been further investigated in PC cell models, where their expressions have been modulated and effects on protein expression evaluated. However, no a clear correlation between miR-23c or miR4328 expressions and PC progression is reached at the end of the study. This study presents several limitations, and Authors require to provide further experimental evidence and justifications. Moreover, Authors should be very careful in drawing conclusions. The title should be also modified because misleading (e.g., effects of downregulation, while only upregulation has been experimentally performed).

Major concerns:

1)      In the analysis of to evaluate differentially expressed miRNAs, 3 categories (Benign, PC, Bone mets) are taken in consideration. While in figures 1A and S1 unsupervised hierarchical clustering of miRNAs are shown, biased clustering based on the three groups on DE-miRNAs should be provided. Moreover, the FDR score for those miRNAs is not clear. How Authors justify that in the validation stage (by RT-qPCR) following the screen the two miRNAs (miR-23c, -4328) result downregulated only in bone tissues?

2)      The strategy of the experimental design with the upregulation of miR-23c and miR-4328 in PC cells should be further discussed. Why Authors decided to upregulate these miRNAs instead of downregulate them? Moreover, basal expression levels of the two miRNAs for both PC cellular models should be presented. Finally, overexpressing cells should be better identified (e.g. differently named from parental ones) since some parts in both text and figures are misleading.

3)      How the Authors justify the presence of upregulated proteins after miR-23c or miR-4328 upregulation in the two different cell models? Moreover, Authors should show any overlap in modulated proteins between the two cell models for each miRNA. Is there any common change between the two models? The Authors should make a comment on this matter. Modulation of direct targets after miRNAs overexpression should be further validated in vitro.

4)      Authors investigated proliferative properties and drug response (Enzalutamide and Simvastatin) as further investigations supported by GSEA analysis results. However, the GSEA analysis revealed that also glycolysis and coagulation pathways were affected by miR-23c as well as miR-4328 modulated other gene signatures. Thus, more evidence to better understand the role of these miRNAs in PC cells should be provided.

5)      EVs content has been evaluated in order to better understand a possible remote modulation of these miRNAs to secondary sites. However, the Authors previously proved that these miRNAs are usually downregulated in both PC tissues and bone mets. Which is the rational to propose a modulator role of these miRNAs? The rational should be better justified. Moreover, how Authors justify that miR-4328 is secreted by a cell line but not by the other?

6)      The effect of only miR-23c is evaluated in vivo, however no convincing rational is providing in order to justify why Authors investigated only this miRNA. Moreover, only one cell model, PC3 has been utilised in this experimental design, while the use of an AR-positive cell line would have been beneficial. Since no obvious differences are observed between experimental and control groups, a different experimental approach should be considered and more evidences are needed to link miR-23c to PC bone metastasis progression.

Minor comments:

Lines 21-24: Sentence too generic and broad. Authors should refer only to specific miRNAs that have been experimentally modulated

Line 26: comma is missing

Line 27: replace ‘into’

Line 24 (and further observations): RT-PCR should be substituted with RT-qPCR

Lines 58-59: this sentence should be rephrased.

Line 59 (and further observations): Capital letter of ‘miRNA’ is required at the beginning of the sentence

Line 79: reference should be placed before, in line 78

Line 82: It seems that samples is referring only to bone metastasis. What about the other tissues (prostate cancer, healthy tissues). Please specify if those samples derived from the same patients. This part of 2.1 section is confusing

Line 112: space missing

Line 140: double mark

Lines 208-209: the authors should indicate/perform additional the GSEA statistics (for instance FDR, p-value)

Line 297-298: the seven miRNAs should be named also in the main text

Lines 318-322: the sentence should be rephrased to better explain results

Reference in the text of figure 2c is missing.

In the Legend 2, there is a reference to figure 2d, but the figure is missing. Please provide explanation and correct this.

Lines 345-346: the experimental design is absolutely unclear and no reference to overexpressing cells is present, so it is assumed that parental cells have been screened. Please rephrase this sentence.

Lines 423-424: The results of this manuscript do not support this hypothesis. Please rephrase it.

Round 2

Reviewer 2 Report

Comments to the Author Authors have addressed all critical issues. I recommend the acceptance of the manuscript.

Author Response

The authors would like to thank reviewer no. 2 for approving the acceptance of the revised manuscript.

Reviewer 3 Report

Response to Authors

REV#3: In this manuscript by Järemo and Collaborators, a miRNA screen in healthy, PC and bone metastasis tissues was performed to evaluate possible roles of miRNAs in PC progression. Two miRNAs, miR-23c and miR4328, have been further investigated in PC cell models, where their expressions have been modulated and effects on protein expression evaluated. However, no clear correlation between miR-23c or miR4328 expressions and PC progression is reached at the end of the study. This study presents several limitations, and Authors require to provide further experimental evidence and justifications. Moreover, Authors should be very careful in drawing conclusions. The title should be also modified because misleading (e.g., effects of downregulation, while only upregulation has been experimentally performed).

Author response: We agree that no clear correlation between expression of miRNA-23c and tumor progression were seen in the experimental model system chosen for the in vivo experiment. This is clearly stated in the results part (section 3.4) and now also in the conclusion (lines 573-75). The title has been changed accordingly. Please find our further response below, addressed item by item.

REV#3 response: Although it is stated that ‘no clear correlation between expression of miRNA-23c and tumor progression were seen in the experimental model system chosen for the in vivo experiment’ in the result section, some parts of the manuscript remain misleading. A clear example is in the simple summary, where Authors claim that ‘Downregulation of those miRNAs during the development of bone metastases may lead to loss of inhibitory effects on prostate cancer cells that need to be further explored’. However, Authors did perform some experiments (that also involved animal experimentation) without reaching any significative result, so this should be clearly stated also in the summary and it is suggested to avoid speculations.

REV#3: In the analysis of to evaluate differentially expressed miRNAs, 3 categories (Benign, PC, Bone mets) are taken in consideration. While in figures 1A and S1 unsupervised hierarchical clustering of miRNAs are shown, biased clustering based on the three groups on DE-miRNAs should be provided. Moreover, the FDR score for those miRNAs is not clear. How Authors justify that in the validation stage (by RT-qPCR) following the screen the two miRNAs (miR-23c, -4328) result downregulated only in bone tissues?

Author response: In the DE analysis, significant miRNAs were selected based on p<0.05 in both comparisons; PC vs. benign and CRPC vs. benign, without adjustment for FDR. The miRNAs-23c, -4328, and the other miRNAs in Fig. 1A further showed significant DE between CRPC and PC (p<0.05) and, thus, the following interesting reduction, benign>localized PC>bone metastases. This together with the p-value <0.05 is given in the text (lines 314-17) and in the figure legends of Fig. 1 and S1. The heatmaps in Fig. 1A and S1 are shown for visualization purpose only, and not for selection. No biased cluster analysis was done. As the reviewer points out, we were not able to verify the downregulation of the miRNAs in localized PC compared to benign tissue, as already commented on in the discussion (lines 511-13).

REV#3 response: the fact that FDR was not taken in consideration to threshold significative miRNAs should be clearly stated in the materials and methods section.

REV#3: The strategy of the experimental design with the upregulation of miR-23c and miR-4328 in PC cells should be further discussed. Why Authors decided to upregulate these miRNAs instead of downregulate them? Moreover, basal expression levels of the two miRNAs for both PC cellular models should be presented. Finally, overexpressing cells should be better identified (e.g. differently named from parental ones) since some parts in both text and figures are misleading.

Author response: Our strategy of upregulating the miRNA-23c and -4328 was based on the finding that both miRNAs were found downregulated in clinical bone metastases compared to benign prostate and localized PC. Thus, we hypothesized that miRNA-23c and -4328 could have tumor suppressive effects and our strategy was to functionally evaluate this after forced overexpression of those selected miRNAs in two PC cell lines with low endogenous levels. This strategy is now clearly described at lines 340-46, and the new Suppl. Fig. S2 has been included to show the relatively low levels of miRNA-23c and -4328 in wildtype PC-3 and 22Rv1 cells in comparison to levels in other cell lines of prostatic origin; two normal epithelial cell lines (RWPE1 and PNT1A), one stromal cell line (WPMY1), and another PC cell line (LNCaP). The text and figures have been revised to consistently point out the overexpressing cells in relation to their wildtype (parental) origin and the corresponding NTC cells. All figures now show the parental cell names and we furthermore believe that the overexpression of miRNAs are clearly pointed out.

REV#3 response: statistical analysis in Figure S2 is missing, please add. How do Authors comment the fact that endogenous levels of LNCaP are very different from 22Rv1 and PC-3? Could have been LNCaP a good cellular model to downregulate the 2 miRNAs in order to have another important evidence on the role of the two miRNAs? Downregulation approach should be performed using LNCaP cells.

REV#3: How the Authors justify the presence of upregulated proteins after miR-23c or miR-4328 upregulation in the two different cell models? Moreover, Authors should show any overlap in modulated proteins between the two cell models for each miRNA. Is there any common change between the two models? The Authors should make a comment on this matter. Modulation of direct targets after miRNAs overexpression should be further validated in vitro.

Author response: The upregulated proteins might be indirectly upregulated, as already stated in the discussion. The overlap with the predictions from target scan might be circumstantial, but the reason is not fully known. This is now clearly discussed, lines 520-24. Only one protein (ITIH4) was significantly downregulated in both cell lines following overexpression of miR-4328 and only two proteins (AHSG and APOH) were significantly downregulated in both cell lines following overexpression of miRNA-23c.This is now commented on in the results part (lines 387-90). ITIH4 is a type II acute-phase protein involved in inflammatory responses to trauma, supposed to be expressed only in the liver. Thus, it is possible that the ITIH4 detected in our experiment was derived from the fetal bovine serum supplemented to the cell culture media. Downregulation of AHSG might be of interest for bone metastasis, as it is reported to influence the mineral phase of bone. Also, the downregulation of APOH may be of interest for tumor progression as it is reported to prevent the intrinsic blood coagulation cascade. However, for functional speculations and verifications, we preferred to focus on the results from the GSEA and not on specific proteins. Therefore, the suggested functions of those specific proteins are not commented on in the discussion. Also, due to the limited functional effects seen in the study following miRNA-23c and-4328 overexpression, we currently believe that validation of direct targets after miRNA overexpression is out of the scope of this paper.

REV#3: Authors investigated proliferative properties and drug response (Enzalutamide and Simvastatin) as further investigations supported by GSEA analysis results. However, the GSEA analysis revealed that also glycolysis and coagulation pathways were affected by miR-23c as well as miR-4328 modulated other gene signatures. Thus, more evidence to better understand the role of these miRNAs in PC cells should be provided.

Author response: First, please see the new Fig. S2 showing results from a wound healing assay, performed to evaluate tumor cell migration following miRNA-23c and-4328 overexpression. Migration was studied to monitor the process of “mesenchymal to epithelial transformation”, proposed by GSEA to be affected in both cell lines by either of the studied miRNAs. Also, we did notice the negative enrichment of “glycolysis” after overexpression of miRNA-23c and-4328 in both cell lines. Our interpretation of this was that reduced glycolysis could be seen as an indirect sign of reduced cell growth following miRNA-23c and-4328, as the glycolysis is highly associated with the Warburg effect seen in cancer and the production of both energy (ATP) and building blocks (nucleotides) needed for DNA and RNA synthesis at high proliferation rate. Therefore, relative cell growth was studied based on the CellTiterGlo assay that is based on ATP production. However, as the fuel for the ATP production is not measured in this assay, additional studies might have been warranted, such as using the Seahorse instrument for a more precise measurement of the glycolysis rate. Furthermore, we had plans for analyzing micro-thromboses in tumors grown in mice, but when the animal study did not show any obvious effects on tumor growth or invasion following miRNA-23c overexpression we did not find those studies motivated.

REV#3: EVs content has been evaluated in order to better understand a possible remote modulation of these miRNAs to secondary sites. However, the Authors previously proved that these miRNAs are usually downregulated in both PC tissues and bone mets. Which is the rational to propose a modulator role of these miRNAs? The rational should be better justified. Moreover, how Authors justify that miR-4328 is secreted by a cell line but not by the other?

Author response: Yes, the miRNA-23c and -4328 are proposed to be downregulated in PC and bone metastases. As such they might have tumor suppressive effects that are reduced by their downregulation. Moreover, we were encouraged by a recent publication [Sharma et al, Oncogene 2021] showing that regucalcin promoted dormancy of prostate cancer by inhibiting angiogenesis, and that this effect was at least partly mediated through miRNA-23c secreted in EVs. Therefore, we specifically measured the levels of miRNA-23c in EVs as well as their possible inhibiting effects on tumor angiogenesis. This strategy was already discussed (lines 546-52) and is now also more clearly expressed in the introduction (lines 77-81). Overall, we focused on the large difference between the relative enrichment of miRNA-23c (up to 2000-fold) and -4328 (up to 2.8-fold) in EVs instead of the slight difference seen for miRNA-4328 between the two cell lines.

REV#3 response: The study provide evidence that these miRNAs are only downregulated between PC and bone mets, but the downregulation in PC versus Benign tissue did not reach significance. This can suggest that PC will produce EVs with low miR-23c and miR-4328 content. However, Authors artificially over-expressed these miRNAs in PC cells (opposite from the hypothesis) and they evaluated that, when overexpressed, there is an increase in miR-23c and miR-4328 content in EVs. However, this experiment does not provide any significative information on the possible role of these miRNAs as cell mediators. Authors should conduct functional study with EVs enriched in miR-23c and miR-4328 content. Moreover, Authors should perform EVs experiment after downregulation of miR-23c and miR-4328 in LNCaP cells in order to have a physio-pathological context closer to what observed in PC patients.

REV#3: The effect of only miR-23c is evaluated in vivo, however no convincing rational is providing in order to justify why Authors investigated only this miRNA. Moreover, only one cell model, PC3 has been utilised in this experimental design, while the use of an AR-positive cell line would have been beneficial. Since no obvious differences are observed between experimental and control groups, a different experimental approach should be considered and more evidences are needed to link miR-23c to PC bone metastasis progression.

Author response: We agree that a different experimental approach would have been needed to better study possible effects of miRNA-23c and 4328 on PC bone metastasis. To the best of our knowledge, however, no adequate experimental model system for prostate cancer bone metastasis exists. Instead, and as also explained above, the rationale behind choosing miRNA-23c for evaluation in vivo was the obvious enrichment of this miRNA in EVs and, thus, the possibility for this miRNA to affect processes in the microenvironment via EV secretion. Our intention with growing PC-3 cells overexpressing miRNA-23c in mice was to examine if this would reduce tumor growth by inhibiting angiogenesis. Due to the lack of obvious effects of miRNA-23c on PC-3 growth or angiogenesis in vivo, ethical and economic reasons made us not to move forward with the other cell line.

REV#3 response: Several animal models of PC bone metastasis exist. Moreover, PC3 cells are known to easily metastasise to bone. Authors should provide further evidences of the role of the overexpression of these two miRNAs after intratibial injection in mice to evaluate the tumour suppressive role. Moreover, an experiment with the downregulation of these miRNAs should be performed in a in vivo setting.

REV#3: Lines 21-24: Sentence too generic and broad. Authors should refer only to specific miRNAs that have been experimentally modulated.

Author response: OK, revised

REV#3: Line 26: comma is missing

Author response: OK, revised

REV#3: Line 27: replace ‘into’

Author response: OK, revised

Line 24 (and further observations): RT-PCR should be substituted with RT-qPCR.

Author response: OK, revised

Lines 58-59: this sentence should be rephrased.

Author response: OK, revised  

Line 59 (and further observations): Capital letter of ‘miRNA’ is required at the beginning of the sentence.

Author response: OK, revised

Line 79: reference should be placed before, in line 78.

Author response: OK, revised

Line 82: It seems that samples is referring only to bone metastasis. What about the other tissues (prostate cancer, healthy tissues). Please specify if those samples derived from the same patients. This part of 2.1 section is confusing.

Author response: OK, revised

Line 112: space missing.

Author response: OK, revised

Line 140: double mark.

Author response: OK, revised

Lines 208-209: the authors should indicate/perform additional the GSEA statistics (for instance FDR, p-value).

Author response: OK, revised in text (lines 215-16, 398-405) and in new fig. S4.

Line 297-298: the seven miRNAs should be named also in the main text.

Author response: OK, revised

Lines 318-322: the sentence should be rephrased to better explain results

Author response: The lines does not seem to be correct. We are sorry for not being able to find the questioned sentence.

Reference in the text of figure 2c is missing.

Author response: Fig. 2 and its references in the text have been revised.

In the Legend 2, there is a reference to figure 2d, but the figure is missing. Please provide explanation and correct this

Author response: OK, revised

Lines 345-346: the experimental design is absolutely unclear and no reference to overexpressing cells is present, so it is assumed that parental cells have been screened. Please rephrase this sentence.

Author response: OK, revised (lines 373-75).

Lines 423-424: The results of this manuscript do not support this hypothesis. Please rephrase it.

Author response: The lines do not seem to be correct. We are sorry for not being able to find and revise the questioned section.

Author Response

Please see the attachment and the author response in red.

Round 3

Reviewer 3 Report

My general consideration is that a few, but extremely important, experimental evidences are still missing. Although Authors have now included those experimental designs as suggested future work, some of those should have been performed in order to reach a conclusion of this work. Particularly, it would have been very informative to have an additional in vivo experiment with the intra-tibial injection of PC3 cells in mice. This experiment would have provide the evidence of the possible role of the two miRNAs in metastasis progression within the bone microenviroment in order to clearly state a functional role for those.